# The Impact of Steatotic Liver Disease on Cytokine and Chemokine Kinetics During Sepsis

**DOI:** 10.3390/ijms26052226

**Published:** 2025-03-01

**Authors:** Nina Vrsaljko, Leona Radmanic Matotek, Snjezana Zidovec-Lepej, Adriana Vince, Neven Papic

**Affiliations:** 1Emergency Infectious Diseases Department, University Hospital for Infectious Diseases “Dr. Fran Mihaljević”, 10000 Zagreb, Croatia; nvrsaljko@bfm.hr; 2Department for Immunological and Molecular Diagnostics, University Hospital for Infectious Diseases “Dr. Fran Mihaljević”, 10000 Zagreb, Croatia; lradmanic@bfm.hr (L.R.M.); szidovec@bfm.hr (S.Z.-L.); 3Department for Viral Hepatitis, University Hospital for Infectious Diseases “Dr. Fran Mihaljević”, 10000 Zagreb, Croatia; avince@bfm.hr; 4Department for Infectious Diseases, School of Medicine, University of Zagreb, 10000 Zagreb, Croatia

**Keywords:** sepsis, bacterial infections, cytokines, chemokines, immune response, transforming growth factor beta, interleukin-17, interleukin-33, liver diseases, non-alcoholic fatty liver disease

## Abstract

Metabolic dysfunction-associated steatotic liver disease (MASLD) has recently been linked with sepsis outcomes. However, the immune mechanisms by which MASLD aggravates sepsis severity are unknown. This prospective cohort study aimed to analyze serum cytokine and chemokine kinetics in patients with MASLD and community-acquired sepsis. Out of the 124 patients, 68 (55%) were diagnosed with MASLD. There were no differences in age, sex, comorbidities, baseline sepsis severity, or etiology between the groups. Serum concentrations of 27 cytokines and chemokines on admission and day 5 of hospitalization were analyzed using a multiplex bead-based assay. Patients with MASLD had significantly higher serum concentrations of IL17A, IL-23, IL-33, CXCL10 and TGF-β1. Different cytokine kinetics were observed; patients with MASLD had a decrease in IL-10, IL-23, CXCL10 and TGF-β1, and an increase in IL-33, CXCL5 and CXCL1 on day 5. In the non-MASLD group, there was a decrease in IFN-γ, IL-6, IL-23 and CCL20, and an increase in CCL11 and CXCL5. While TGF-β1 significantly increased in non-MASLD, in MASLD, it decreased on day 5. Kinetics of TGF- β1 and CCL11 were associated with mortality in patients with MASLD. In conclusion, MASLD is linked with distinct cytokine and chemokine profiles during sepsis.

## 1. Introduction

Sepsis is defined as a potentially life-threatening organ dysfunction resulting from a dysregulated immune response to infection [1]. This definition highlights the pivotal role of innate and adaptive immune responses and suggests the loss of pro-inflammatory and anti-inflammatory balance leading to tissue damage and impaired bacterial clearance. This excessive inflammation is orchestrated by a plethora of immune response mediators, including cytokines and chemokines, as well as acute phase response proteins mainly produced in the liver [2,3]. Although significant progress has been made in understanding the pathophysiology of sepsis, no immune response-directed therapies showed clinical efficacy. This is repeatedly explained by sepsis complexity and heterogeneity where systemic inflammatory response is dependent on the culprit pathogen, site of the infection and multiple patients’ factors, including comorbidities, obesity or metabolic conditions.

Metabolic dysfunction-associated steatotic liver disease (MASLD) is the most common cause of chronic liver disease, with a rising prevalence in the general population (>35%) [4,5]. MASLD encompasses a spectrum of liver conditions, extending from most common simple steatosis to steatohepatitis and cirrhosis [4,5]. Currently, MASLD is considered a systemic disease characterized by chronic low-grade inflammation accompanied by altered immune responses, which play a central role in progression of liver disease and development of extrahepatic complications [6].

There is growing evidence that MASLD is associated with infection risk, severity and outcomes, as in community-acquired pneumonia [7,8], COVID-19 [9,10], bacteriemia originating from the digestive tract [11], *Clostridioides difficile*-associated disease [12,13] and urinary tract infection [14]. More recently, in several cohorts, MASLD was linked with sepsis severity and outcomes, including higher frequency of liver and kidney injury, progression to septic shock and higher mortality [15,16,17].

However, the reason for the increased susceptibility and mechanisms by which MASLD aggravates bacterial infections are still unknown. The hypothesis that patients with MASLD might have different sepsis immune phenotypes has not been explored, and the characteristics of the immune response in this patient group have not been described.

The aim of this prospective observational study was to analyze the profile and kinetics of serum concentrations of inflammatory cytokines and chemokines in patients with sepsis, depending on the presence of MASLD, and to assess their impact on clinical outcomes.

## 2. Results

The cohort consisted of 124 hospitalized adult patients (56 (46%) males) with community-acquired sepsis, with a median age of 65 (IQR 54–74). The patients were divided into two groups according to the presence of criteria for MASLD: patients diagnosed with MASLD (68, 55%) and non-MASLD group (56, 45%).

### 2.1. Baseline Patients’ Characteristics

As presented in Table 1, there were no differences in age, sex and comorbidities between the two groups. The most common comorbidities detected in our cohort were arterial hypertension, type 2 diabetes mellitus (T2DM), dyslipidemia and cardiovascular diseases (Table 1). Patients with MASLD were more frequently obese with a higher BMI (32, IQR 27–39 kg/m^2^ vs. 25, IQR 22–28 kg/m^2^) and waist–hip ratio (WHR; 1.0, IQR 0.97 vs. 0.98, IQR 0.95–1, *p* < 0.01) and lower waist–height ratio (WHtR; 1.6, IQR 1.5–1.8 vs. 1.9 IQR 1.8–2.1, *p* < 0.01). Patients with MASLD more frequently fulfilled criteria for metabolic syndrome (MetS) (43, 63% vs. 19, 34%, *p* = 0.002).

The median controlled attenuation parameter (CAP) in patients with MASLD was 302 db/m (IQR 271–360), and in patients without MASLD, it was 193 db/m (IQR 162–210); 38 (56%) patients had grade 3 steatosis, 19 (28%) had grade 2 and 11 (16%) had grade 1 steatosis. FAST score, as a surrogate marker of liver damage in this setting was significantly higher in patients with MASLD (0.43, IQR 0.16–0.64 vs. 0.16, IQR 0.05–0.42, *p* < 0.01).

The most common source of sepsis in both groups was lower respiratory tract, followed by skin, urinary tract and gastrointestinal source. The pathogen was detected in blood culture in a total of 44 patients (35%), urine culture in 26 (21%), respiratory samples in 10 (8%), stool in 10 (8%) and other samples (swabs, aspirates or punctate) in 17 (14%). The most common isolates in blood cultures were *E. coli* (14, 11%), *S. aureus* MSSA (11, 9%), *S. pneumoniae* (8, 6%) and *K. pneumoniae* (5, 4%). There were no differences in the infection source or etiology between the groups.

Time from symptom onset to hospital admission was similar (3, IQR 2–6 vs. 4, IQR 2–7 days, *p* = 0.276), and most patients in both groups had moderate disease severity on admission as measured by SOFA and APACHE II scores (Table 1).

Laboratory findings on admission are shown in Table 2. As presented, patients with sepsis frequently had elevated inflammatory markers C-reactive protein, procalcitonin, and fibrinogen, and leukocytosis with neutrophilia. There were no significant differences in routine laboratory markers, except for GGT and triglycerides, which were higher in the MASLD group.

### 2.2. Analysis of Cytokine and Chemokine Profiles in Septic Patients with MASLD

We examined if there are any differences in serum concentrations of selected cytokines and chemokines on day 1 and 5 of hospitalization between the two groups. As presented in Figure 1 and Figure 2, upon admission, patients with MASLD had higher concentrations of IL17A, IL-23, IL-33, CXCL10 and TGF-β1. On the 5th day of hospitalization, the MASLD group had higher concentrations of IFN-γ, IL-17A, IL-33, CCL17, CCL20 and CXCL1. There was no difference in the concentrations in remaining cytokines and chemokines (complete data are provided in Appendix A).

Next, we analyzed the differences in kinetics of measured cytokines/chemokines between the two groups and observed different patterns. The MASLD group had a decrease in IL-10, IL-23, CXCL10 and TGF-β1, while in the non-MASLD group, there was a decrease in IFN-γ, IL-6, IL-23 and CCL20 (Figure 1 and Figure 2). Patients with MASLD had significant increases in IL-33, CXCL5 and CXCL1, while in the non-MASLD group, CCL11 and CXCL5 significantly increased on day 5. Of note, the MASLD group had higher TGF-β1 serum concentrations upon admission, but these levels did not further increase as in the non-MASLD group but decreased during a second phase of sepsis.

Regarding routine laboratory markers, there was a significant decrease in CRP and procalcitonin on day 5 in the non-MASLD group, but not in the MASLD group (Appendix A). On day 5, CRP was significantly higher in the MASLD group than the non-MASLD group. WBC and hemoglobin significantly decreased in both groups, while neutrophils-to-lymphocytes ratio and platelet count significantly increased. At hospital discharge, patients in the non-MASLD group had significantly higher platelet count (Appendix A).

A correlation analysis was performed to identify correlations of measured cytokines and chemokines with steatosis grade (measured by CAP), FAST score and BMI. CAP positively correlated with IFN-γ (r = 0.16) and TARC (r = 0.23) on day 5, and IL-23 (r = 0.26) on day 1; and with IL17A (D1 r = 0.25 and D5 r = 0.39), IL-33 (r = 0.3 and r = 0.34) and CXCL10 (D1 r = 0.24, D5 r = 0.2) on day 1 and 5. While CAP positively correlated with TGF-β1 (r = 0.18) on day 1, it showed negative correlation on day 5 (r = −0.16). FAST score, which takes into account liver steatosis, liver stiffness and AST levels [18], showed a positive correlation with TNFα on day 1 and 5 (r = 0.22 and r = 0.18, respectively), IL-18 (r = 0.19) and CXCL10 on day 1 (r = 0.17), and a negative correlation with TGF-β1 on day 5 (r = −0.19). Complete data on correlation analysis are provided in Appendix A.

### 2.3. Association of Cytokine and Chemokine Serum Concentrations with Sepsis Severity in Patients with and Without MASLD

Further, a correlation analysis was performed to investigate the association of cytokines and chemokines with sepsis severity, as measured by SOFA score (Table 3). A significant difference was found in patients with MASLD as compared to the non-MASLD group. While patients with MASLD showed a positive correlation with INF-α2 on day 1 and 5 (r = 0.30 and r = 0.23, respectively), a negative correlation with IFN-γ was found in the non-MASLD group (r = −25, r = −0.37). IL-6 positively correlated with SOFA score in the MASLD group (r = 0.25, r = 0.37), but not in the non-MASLD group. IL-10 on day 1 showed a positive correlation in the MASLD group (r = 0.25) but was negative on day 5 in the non-MASLD group (r = −0.34). IL-8 and IL-18 on day 1 (r = 0.26 and r = 0.24) and IL-1β on day 5 (r = −0.45) showed correlation in the non-MASLD group, but not in the MASLD group, where TGF-β1 negatively correlated with sepsis severity on day 5 (r = −0.22). IL-23 showed a strong negative correlation on day 5 in both groups (r = −0.31 and r = −0.44). IL-33 had a negative correlation at both time points (r = −0.23 and r = −0.33) in patients with MASLD.

Regarding chemokines, in the MASLD group, correlation with eotaxin (r = −0.27), ENA-78 (r = −0.23), I-TAC (r = −0.21) and MIP-1β (r = 0.35) was identified, while in the non-MASLD group, correlation with eotaxin (r = −0.24), TARC (r = −0.26) and ENA-78 (r = −0.40) was identified. Details on correlation analysis are presented in Table 3.

In addition, we examined the correlation of SOFA score with liver steatosis, liver stiffness and liver-related scores (FAST, APRI and FIB-4) in patients with MASLD. As shown in Appendix A, SOFA correlated with liver steatosis grade (CAP, r = 0.2, *p* = 0.02) and FAST score (r = 0.3, *p* < 0.01), as well as with APRI (r = 0.44, *p* < 0.01) and FIB-4 (r = 0.55, *p* < 0.01).

### 2.4. Analysis of Cytokine and Chemokine Kinetics with Sepsis Survival in Patients with MASLD

Next, we examined if there is an impact of MASLD on cytokine and chemokine kinetics in survivors vs. non-survivors. Overall, 24 (19.4%) patients in our cohort died during hospitalization: 18 (25%) with MASLD and 7 (12.5%) in the non-MASLD group. Serum concentrations of sepsis survivors vs. non-survivors stratified by the presence of MASLD are shown in Appendix A. A series of three-way RM-ANOVA analyses was performed, which identified four biomediators significantly associated with MASLD and mortality in our cohort: TGF-β1, IFN-γ, CXCL10 (IP10) and Eotaxin, as presented in Figure 3.

In a subsequent ROC analysis aimed to identify cut-off values for predicting mortality in patients with MASLD and non-MASLD, TGF-β1 showed a moderate predicting role in both groups, but with different and opposite cut-off values (Figure 4). In patients with MASLD, TGF-β1 > 230 pg/mL showed sensitivity of 67% (95%CI 50–86%) and 71% (95%CI 58–81%) with an AUC of 0.73 (95%CI 0.57–0.88). In contrast, in patients with non-MASLD, decreased TGF-β1 < 60 pg/mL concentrations had a sensitivity of 78% (95%CI 50–95%) and specificity of 67% (95%CI 52–76%) with an AUC of 0.71 (95%CI 0.55–0.86). Serum concentrations of Eotaxin on day 5 had an AUC of 0.71 (95%CI 0.55–0.88) in MASLD and 0.75 (95%CI 0.58–0.91) in the non-MASLD group (Figure 4). However, in the MASLD group, Eotaxin < 140 pg/mL had a sensitivity of 66% (95%CI 40–85%) and specificity of 62% (95%CI 52–75%), while in the non-MASLD group, increased Eotaxin > 215 pg/mL showed a sensitivity of 75% (95%CI 46–93%) and specificity of 70% (95%CI 58–79%).

Finally, we performed a survival analysis which showed the association of increased serum concentrations of TGF-β1 > 230 pg/mL on day 1 with mortality in the MASLD group (HR 4.6, 95%CI 1.6–14, *p* = 0.0052), but not the non-MASLD group (Figure 5). In contrast, decreased levels of TGF-β1 < 60 pg/mL were associated with mortality in the non-MASLD group (HR 7.8, 95%CI 1.2–49, *p* = 0.030). Similarly, Eotaxin on day 5 < 140 pg/mL was associated with mortality in the MASLD group (HR 3.6, 95%CI 1.1–11, *p* = 0.03), but not in the non-MASLD group.

## 3. Discussion

Here, we provide the first evidence that sepsis patients with MASLD have distinct serum cytokine and chemokine profiles compared to patients without MASLD. This includes higher concentrations of Th-17 interleukins (IL17A, IL-23), anti-inflammatory Th2 interleukin IL-33, immunomodulatory TGF-β1, IFN-γ and several chemokines, including CXCL10, CCL17, CCL20 and CXCL1. There were no differences in inflammatory IL-6, IL-8 or anti-inflammatory IL-10 levels. Notably, patients with MASLD had different serum kinetics of IL-6, IL-10, IL-23, IL-33, TGF-β1 and IFN-γ, as well as of chemokines CXCL10, CCL11, CCL17, CCL20 and CXCL1.

Surprisingly, although cytokine and chemokine responses in sepsis have been extensively studied, to date, there are no reports on the characteristics of inflammatory responses in patients with MASLD. That patients with MASLD might have different immune responses was shown in COVID-19, where several cytokines and chemokines were linked with uncontrolled inflammation and development of severe/critical disease (such as IL-6, IL-8, IL-10, TGF-β1 and CXCL10) [19,20]. Studies analyzing sepsis cytokine responses associated with other components of metabolic syndrome, mainly obesity and diabetes mellitus, reported disagreeing results. While some showed no significant impact [21], others showed significant perturbations in inflammatory markers in patients with T2DM [22,23]. Similarly, obesity was linked with higher TNF-α and oxidative stress markers [24,25], adipocytokines [26] and increased IL-6, IL-8 and IL-10 in some studies [27]. However, none of them included MASLD as a variable or examined its impact on cytokine profiles.

The immune response in sepsis has been simplified in two subsequent phases: a cytokine-mediated hyper-inflammatory phase characterized by increased release of pro-inflammatory mediators (such as TNF-α, IL-1, IL-12 and IL-6), which might lead to “cytokine storm” and serious multi-system dysfunction; and a second phase of compensatory anti-inflammatory response, which is facilitated through the release of anti-inflammatory cytokines, such as TGF-β, IL-4 and IL-10, to restore homeostasis [28,29]. However, if the hypoinflammatory phase is prolonged, it can lead to significant immunosuppression and increase susceptibility to secondary infections [28,29,30].

While we identified extensive changes in cytokine profiles, several findings are to be highlighted.

First, patients with MASLD exhibited higher levels of inflammatory cytokines linked with Th17 responses, IL-17, IL-23 and IL-33. IL-23 and IL-17 are closely related pro-inflammatory cytokines that play a significant role in the development and progression of chronic inflammation, including liver fibrosis and MASLD [31,32]. In the context of chronic inflammation, antigen-stimulated dendritic cells and macrophages produce IL-23, which stimulates the expansion of Th17 cells. Production of IL-17 activates neutrophils and triggers strong inflammatory responses by amplifying cytokine production while suppressing the Treg cell development [31,32]. This imbalance leads to the hyperactivation of the Th17/IL-17 system. In the liver, this sustains ongoing inflammation and fibrinogenesis in the liver. In sepsis, several reports described an important IL-17 role: elevated serum IL-17 were shown to increase the susceptibility for septic complications in polytrauma patients [33]; increased tissue and plasma levels were linked with multiorgan failure [34], specifically development of ARDS [35] and acute kidney injury [36]. Recently, it was shown that sepsis induces chronic non-specific production of IL-17 by CD4 T cells, resulting in the inability to mount an effective immune response and increasing the susceptibility to secondary infections in sepsis survivals [37].

IL-33 is a cytokine from the IL-1 family that modulates Th2 responses and inhibits T- cell differentiation into Th17 cells [38,39]. Higher concentrations of IL-33 have been found in various liver diseases, such as MASLD, viral hepatitis and cirrhosis [40]. In chronic liver diseases, increased expression of IL-33 enhances Th2 responses, activates Stellate cells and promotes fibrinogenesis [40]. During early sepsis, IL-33 plays an important anti-inflammatory role, enhances neutrophil migration and bacterial clearance, decreases lymphocyte apoptosis and suppresses Th17 responses, which improves early survival [38,39]. However, high levels can lead to immune suppression in sepsis survivors by promoting Treg expansion through IL-10 production, leading to chronic immune dysfunction [39,41].

We can hypothesize that increases in IL-17, IL-23 and IL-33 in septic patients with MASLD might explain frequent sepsis complications previously reported in other cohorts, but also long-term immunological dysfunction in sepsis survivors, which was not examined here but might facilitate subsequent liver disease progression.

Next, we report different kinetics of serum TGF-β concentrations in patients with MASLD. Upon admission, patients with MASLD had higher serum TGF-β1 concentrations in the early phase of sepsis but failed to further increase, as in the non-MASLD group, during the compensatory anti-inflammatory phase of sepsis. In MASLD patients, TGF-β1 concentrations significantly decreased. Furthermore, increased levels of TGF-β1 on admission were a predictor of mortality in patients with MASLD, but decreased levels were a predictor in the non-MASLD group. TGF-β is a pleiotropic cytokine that plays a critical role in cellular communication but also has a dual role in the immune response; while it drives inflammation by promoting the differentiation of Th17 and Th22 cells, its primary function is anti-inflammatory, as it inhibits effector immune cells, including T cells, B cells, dendritic cells and the cytotoxic activity of NK cells [42]. Additionally, TGF-β1 has an inhibitory effect on the maturation of macrophages and blocks the production of cytokines making it key in suppressing immune responses and preventing uncontrolled inflammation, as extensively reviewed in [42]. Dysregulation of the TGF-β pathway in MASLD drives fibrinogenesis by increasing the production of extracellular matrix proteins and is one of the key pathways regulating this process [43].

In sepsis, data on the predictive role of TGF-β are scarce and often conflicting. In severe community-acquired pneumonia, decreased TGF-β1 levels were predictors of mortality [44], while in sepsis-induced ARDS, increased levels of TGF-β1 were associated with progressive disease and fatal outcomes [45]. In burn patients, a low secondary TGF-β1 response was associated with sepsis mortality [46]. In a cohort of critically ill children with sepsis, TGF-β1 did not correlate with sepsis severity or outcomes [47]. Recently, the importance of the TGF-β1 pathway for sepsis susceptibility was recently shown in single-nucleotide polymorphism analysis which identified SNPs in the TGF-β1 gene associated with increased baseline and higher risk for sepsis development [48]. In contrast, in a small cohort study, sepsis survivors had lower admission levels of TGF-β1, which did not significantly differ during sepsis between survivors and non-survivors [49]. Of note, none of these studies examined the impact of MASLD or metabolic syndrome on TGF-β1 concentrations. In COVID-19, we examined TGF-β1 serum concentrations in one point at hospital admission in patients with MASLD and reported that admission TGF-β1 levels were significantly higher in MASLD patients and correlated with disease severity, severe ARDS, time to recovery and mortality [20], as reported here in patients with sepsis. The possible pathophysiological explanations and clinical implications of distinct TGF-β1 kinetics in patients with MASLD are unclear and should further be explored.

IFNγ primes chemotaxis of neutrophils at the site of inflammation and phagocytosis by neutrophils. Once produced, it stimulates the production and secretion of IFNγ-induced protein-10 (IP-10 or CXCL10), which further primes chemotaxis and neutrophil phagocytosis [50]. More recently, a novel immunosuppressive effect of IFN-γ and mTOR signaling in septic animal models was described; during sepsis, activated NK T-cells stimulate secretion of IFN-γ by NK cells in a mechanism mediated by mTOR resulting in impairment of macrophage phagocytic function [51]. Inhibition of mTORC during sepsis decreases IFN-γ secretion by NK cells, normalizes phagocytic function of macrophages and improves mice survival of secondary candidemia [51]. Likewise, it was shown that a decrease in circulating concentrations of IFNγ over the first 72 h was linked with favorable outcomes in humans [52]. Interestingly, we found a significant decrease in IFNγ in non-MASLD patients, but not in MASLD patients, whereas MASLD patients had significantly higher serum concentrations on day 5 of hospitalization. Furthermore, we increased serum concentrations of CXCL10 (IP-10) in MASLD patients on admission. High CXCL10 levels have been linked with the development of acute kidney injury in pediatric sepsis patients [53], progression to septic shock [54], ARDS and lung fibrosis in COVID-19 [19,55]. In mice models, better survival with less bacterial burden in the lungs and blood was shown in CXCL10-deficient mice, suggesting that CXCL10-targeted treatments might be promising approaches for acute sepsis [56]. Both CXCL10 and IFNγ are increased in patients with MASLD and linked with development of steatohepatitis and fibrosis [57,58], which could partially explain their increased levels in septic patients. Furthermore, while IFNγ showed a strong negative correlation with SOFA score in the non-MASLD group, IFNα in patients with MASLD showed a strong positive correlation, suggesting different IFN regulation. This could be explained by a finding that obese patients with MASLD have increased IFNα and decreased IFNγ levels, and IFNα but not IFNγ levels are associated with the accumulation of intramuscular fat, an important contributor to insulin resistance [59].

We also found significant differences in several other chemokines, including CCL17, CCL20 and CXCL1, which were all significantly higher in the MASLD group on day 5. Furthermore, CC17 and CXCL1 significantly increased in MASLD patients, while remaining unchanged in non-MASLD patients. In contrast, CCL20 significantly decreased in non-MASLD patients. While chemokines are considered key regulators of local inflammation, their role in sepsis has not been extensively studied, and their function is mainly extrapolated from other conditions or animal models. CCL17 (or TARC) was shown to correlate with the severity of various autoimmune and inflammatory diseases [60]. In a mice sepsis model, it was suggested that CCL17 through its receptor CCR4+ on Tregs has prolonged immunosuppressive effects [61]. CXCL1 was linked with IL-17 production and Th17 cell regulation, with an important role in neutrophil recruitment and bacterial clearance in a mice polymicrobial sepsis [62] and *S. pneumonie*-derived sepsis model [63]. CCL20 was reported to be significantly higher in septic patients in surgical ICU and correlated with SOFA score [64]. Interestingly, while CCL11 (eotaxin) significantly increased in non-MASLD patients, decreased levels were associated with mortality in the MASLD group. In sepsis-induced myocardial injury patients, CCL11 was associated with severity and mortality [65]. Similarly, in a cohort of 80 patients admitted to the ICU, CCL11 correlated with SOFA score [66]. Whereas CCL11 is mainly involved in the regulation of allergic responses and eosinophil recruitment, recently it was implicated in the progression of MASLD [67]. Our results emphasize the possible effects of MASLD on chemokine responses in sepsis that warrants further explorations.

Our findings should be viewed within the study’s limitations. Of note, this study was not designed to examine the impact of MASLD on sepsis complications or survival, but to detect general differences in cytokine/chemokine expression in septic patients with MASLD. This restricted analysis of cytokine kinetics in patients with different sepsis subphenotypes, specific organ damage, different etiologies and treatment regiments. We focused exclusively on patients with community-acquired bacterial infections, and these results may not reflect the impact of MASLD in other clinical contexts. While the groups were well balanced regarding comorbidities, source and etiology of sepsis, additional confounding could not be excluded. Since sepsis is a heterogeneous syndrome with multiple factors influencing its outcomes, a relatively small number of participants in our cohort limits the statistical subgroup analysis, including the association of specific cytokines with survival, and should be confirmed in larger multicentric studies. Concentrations of cytokines were analyzed at two time points, and there was no long-term follow-up.

Nonetheless, we studied a relatively large and well-defined cohort of patients, providing the first description of the unique characteristics and changes in the cytokine and chemokine expressions among MASLD patients with sepsis. Additional research investigating complex and aberrant immune responses in MASLD is needed.

## 4. Materials and Methods

### 4.1. Study Design and Population

A prospective, non-interventional monocentric observational study was conducted at the University Hospital for Infectious Diseases (UHID) Zagreb, Croatia (SepsisFAT, ClinicalTrials.gov Identifier: NCT06021743). Included were patients with a diagnosis of bacterial sepsis according to the Sepsis-3 criteria [1] with SOFA score ≥ 2 and clinical suspicion of bacterial infection acquired in the general population. Predefined exclusion criteria were as follows: age <18 years; if more than 24 h have passed since hospitalization; pregnancy; immunocompromised patients (including HIV, malignant diseases, immunosuppressive therapy); chronic viral hepatitis; other chronic liver diseases; excessive alcohol consumption (>20 g of alcohol per day for women and >30 g for men [4]); taking hepatotoxic drugs; infection of the central nervous system.

All participants gave written informed consent. The study conformed to the ethical guidelines of the Declaration of Helsinki and was approved by the Ethics Committee of the UHID Zagreb.

### 4.2. Data Collection and Definitions

Upon admission, we collected demographic information (age, sex), comorbidity data (metabolic syndrome components, cardiovascular, kidney, and neurological conditions), anthropometric data (body mass index (BMI), waist circumference (WC), waist–hip ratio (WHR), and waist–height ratio (WHtR)), chronic medications, symptoms, baseline clinical status, and origin/clinical syndrome of infection. Disease severity upon admission was assessed using APACHE II and SOFA scores. Routine laboratory workup at admission and on the 5th day of hospitalization, including C-reactive protein (CRP), procalcitonin (PCT), lactate, urea, creatinine, bilirubin, aspartate aminotransferase (AST), alanine aminotransferase (ALT), gamma-glutamyl transferase (GGT), alkaline phosphatase (ALP), lactate dehydrogenase (LDH), white-blood-cell count (WBC), neutrophils, lymphocytes, hemoglobin, platelets, glucose, coagulation tests, total proteins, and albumin, were collected, as well as a lipid panel (triglycerides, cholesterol, high-density lipoprotein (HDL), and low-density lipoprotein (LDL)), taken in the morning and on an empty stomach. According to study protocol, hepatitis B and C, and HIV serology was performed in all patients. Minimal microbiology workup included 2 sets of blood cultures, urine culture, and additional microbiological samples based on the clinical syndrome. On admission and on the 5th day of hospitalization, we obtained an additional serum sample for the analysis of cytokine and chemokine concentrations.

Within 48 h of admission, the presence and degree of liver steatosis were determined using the controlled attenuation parameter (CAP). The following cut-off values were used: CAP: <248 dB/m (S0, no steatosis), ≥248 dB/m (S1, mild steatosis), >280 dB/m (S2, moderate steatosis), and >300 dB/m (S3, severe steatosis) [68]. Patients were subsequently diagnosed with MASLD according to current guidelines that require liver steatosis in the absence of other causes of liver diseases with the presence of at least one of the four metabolic risk factors: (1) BMI ≥ 25 kg/m^2^ or waist circumference > 94 cm for men and 80 cm for women; (2) glucose intolerance or type 2 diabetes; (3) arterial hypertension; (4) dyslipidemia [68]. The stage of liver disease was also assessed by calculating APRI, FIB-4 and FAST scores [18,69].

Patients were monitored daily until discharge, and clinical course and outcomes of the disease were recorded.

### 4.3. Cytokine and Chemokine Measurement in the Patients’ Sera

For the analysis of serum concentrations of preselected cytokines and chemokines, the “flow cytometer microsphere-based assay” (FMBA) multiplex test method was used. We used three predefined LEGENDplex panels (Biolegend, San Diego, CA, USA) that include the following: (1) LEGENDplex™ Human Inflammation Panel 1 (13-plex) for detection of IL-1β, IFN-α2, IFN-γ, TNF-α, MCP-1, IL-6, IL-8, IL-10, IL-12p70, IL-17A, IL-18, IL-23, IL-33; (2) LEGENDplex™ HU Proinflammatory Chemokine Panel 1 (13-plex) for detection of MCP-1 (CCL2), IP-10 (CXCL10), Eotaxin (CCL11), TARC (CCL17), MIP-1α (CCL3), MIP-1β (CCL4), MIG (CXCL9), MIP-3α (CCL20), ENA-78 (CXCL5), GROα (CXCL1), I-TAC (CXCL11), IL-8 (CXCL8) and (3) LEGENDplex™ Human Free Active/Total TGF-β1 Assay for detection of free TGF-β1. A flow cytometer BD FACSCanto II (Beckton Dickinson, Franklin Lakes, NJ, USA) was used for detection, and the concentrations of analyzed cytokines and growth factors were analyzed in the LEGENDplex software (version 8.0., Biolegend, San Diego, CA, USA). These panels have been validated by both the manufacturer (Biolegend, San Diego, CA, USA) and our laboratory. They have undergone rigorous testing and have been extensively utilized in various research publications, including those authored by scientists from our group.

### 4.4. Statistical Analysis

The data were evaluated and descriptively presented as frequencies and medians with interquartile ranges (IQRs). The Shapiro–Wilk test was used to check if a continuous variable followed a normal distribution. The cohort was stratified depending on the fibroelastography findings into MASLD and non-MASLD groups, as previously described, and Fisher’s exact test and the Mann–Whitney U test were used to compare the groups. The correlation of cytokine and chemokine concentration with other laboratory parameters was determined by Spearman’s correlation analysis. The Wilcoxon matched pairs test was used to analyze the changes in concentrations at baseline and at day 5 within the groups. A two-way repeated measures ANOVA test with Tukey’s multiple comparisons test (the *p*-values were not adjusted for multiplicity) was used to analyze the comparison of cytokine and chemokine concentrations between the two groups at two measurement points, and RM three-way ANOVA was used for the analysis of whether any of three variables (liver steatosis, survival and time) influenced cytokine/chemokine kinetics, and the relationship between them. The sample size of at least 120 participants was selected according to power analysis for the RM-ANOVA test to achieve an 80% chance of detecting a difference in cytokine concentrations at a 5% significance level. ROC analysis was used to determine the threshold values of serum concentrations of cytokines and chemokines, and their predictive values for the sepsis mortality were evaluated with survival analysis using the Kaplan–Meier method, and the comparison between groups was made using the log-rank test. Statistical analyses were performed using GraphPad Prism Software version 10 (San Diego, CA, USA).

## 5. Conclusions

In conclusion, we present the first evidence that patients with MASLD have different cytokine and chemokine responses during community-acquired bacterial sepsis, including increase in biomediators which have already been implicated in MASLD progression, IL-17, IL-23, IL-33, CXCL10 and TGF-β1, and linked with early sepsis outcomes and prolonged immunosuppressive state in sepsis survivals. Since personalized treatment approaches directed to unique host-immunological characteristics are increasingly being explored, a better understanding of immune responses to bacterial infections in MASLD patients might have significant implications for future research on novel prognostic and therapeutic strategies.

## Figures and Tables

**Figure 1 ijms-26-02226-f001:**
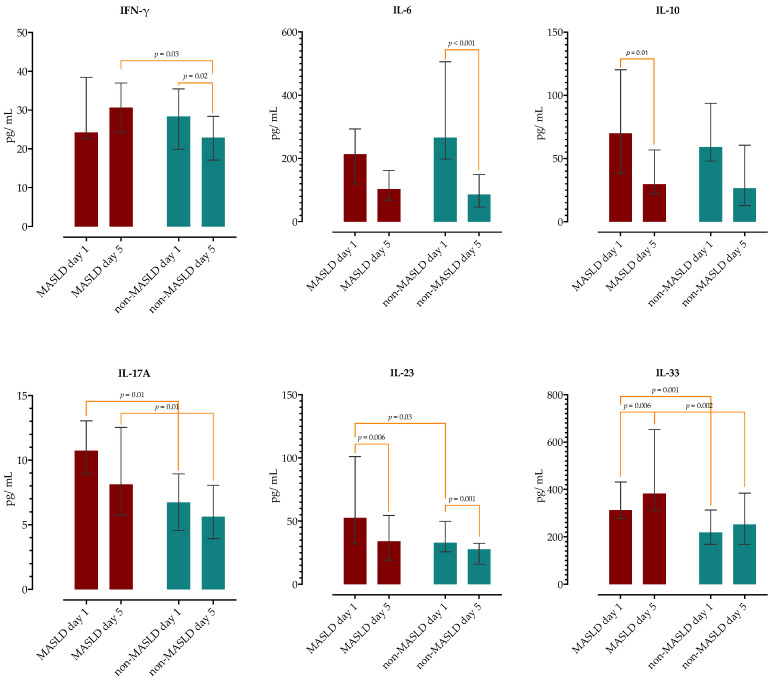
Analysis of serum cytokine kinetics on day 1 and 5 of hospitalization in patients with and without MASLD. The height of the bars represents medians and whiskers IQRs. Wilcoxon rank sum test was performed to analyze the difference in time within the groups and Mann–Whitney test between the groups.

**Figure 2 ijms-26-02226-f002:**
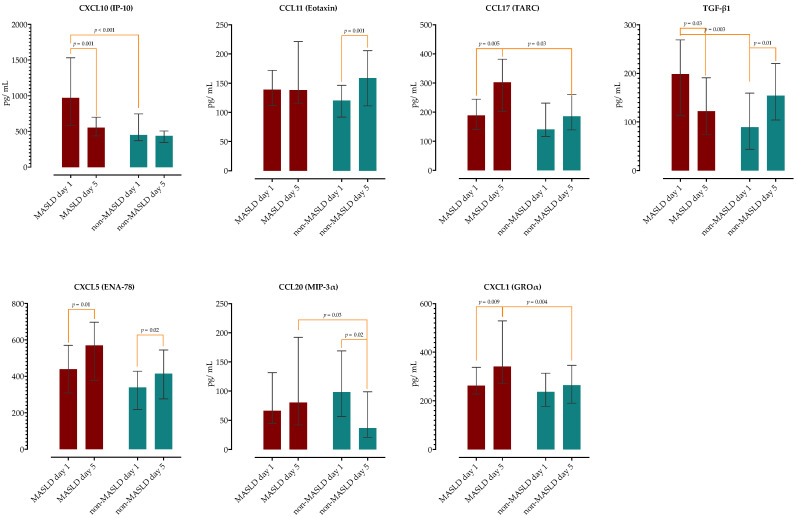
Analysis of serum chemokine kinetics on day 1 and 5 of hospitalization in patients with and without MASLD. The height of the bars represents medians and whiskers IQRs. Wilcoxon rank sum test was performed to analyze the difference in time within the groups and Mann–Whitney test between the groups.

**Figure 3 ijms-26-02226-f003:**
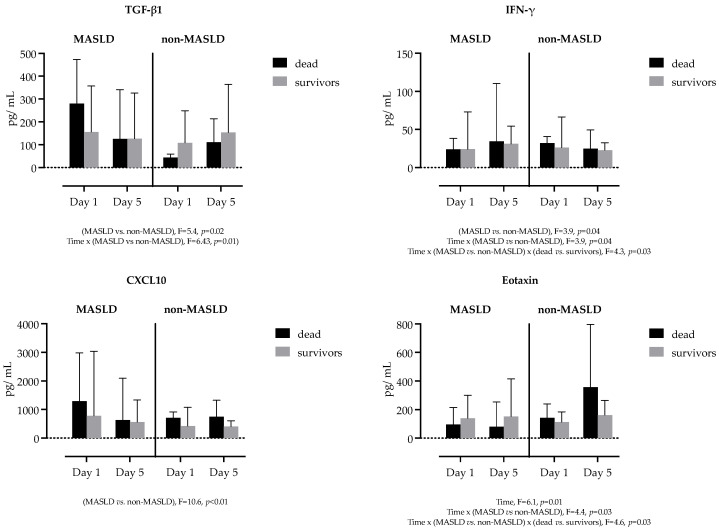
Comparison of TGF-β1, IFN-γ, CXCL10 (IP10) and Eotaxin serum concentrations at two selected time points, stratified by the presence of MASLD and sepsis outcome (survivors vs. non-survivors). The height of the bar represents median concentration, and the extending whisker is the 75% percentile. Repeated measures three-way ANOVA with Tukey’s multiple comparisons test was used to calculate the source of variations. The reported *p*-values have not been adjusted for multiplicity. Show are *p*-values: *p* < 0.05 is considered significant.

**Figure 4 ijms-26-02226-f004:**
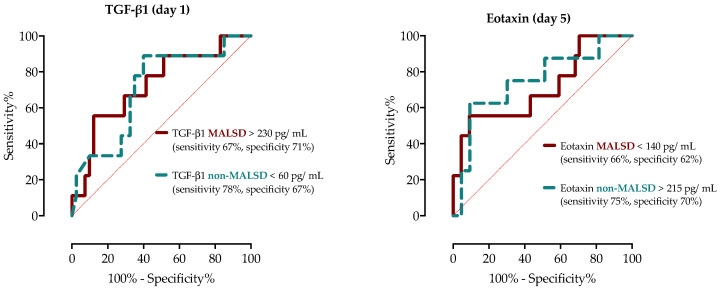
ROC analysis of TGF-β1 serum concentrations on day 1 and Eotaxin on day 5 in patients with and without MASLD with corresponding cut-off values, sensitivity and specificity.

**Figure 5 ijms-26-02226-f005:**
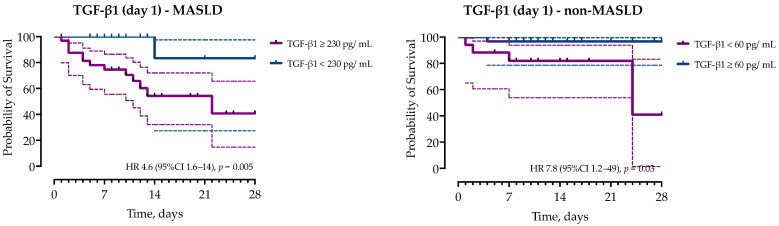
Kaplan–Meier curves with error bands showed in dashed lines and hazard ratios (HRs) with corresponding 95% confidence intervals (95% CI) for the probability of 28-day survival stratified by the concentrations of TGF-β1 on day 1 in patients with and without MASLD.

**Table 1 ijms-26-02226-t001:** Baseline patients’ characteristics.

	MASLD (*n* = 68)	Non-MASLD (*n* = 56)	*p*-Value
Age, median (IQR)	65 (58–74)	64 (46–74)	0.461
Male, No. (%)	31 (46%)	25 (45%)	1.000
BMI (kg/m^2^)	32 (27–39)	25 (22–28)	<0.001
CAP, db/m	302 (271–360)	193 (162–210)	<0.001
Liver stiffness, kPa	6.6 (5.5–11)	6.2 (4.2–9.6)	0.140
Comorbidities			
Diabetes Mellitus	21 (31%)	11 (20%)	0.216
Dyslipidemia	18 (26%)	10 (18%)	0.286
Arterial Hypertension	43 (63%)	26 (46%)	0.071
COPD	6 (9%)	2 (4%)	0.292
Gastritis/GERD	3 (4%)	7 (12%)	0.183
Cardiovascular Disease	19 (28%)	15 (27%)	1.000
Chronic Kidney Disease	5 (7%)	3 (5%)	0.728
Peripheral Vascular Disease	5 (7%)	8 (14%)	0.248
Neurological disease	7 (10%)	8 (14%)	0.584
Smoking	12 (18%)	14 (25%)	0.378
Source of sepsis			
Respiratory Tract	24 (35%)	23 (41.07%)	0.0853
Urinary Tract	11 (16%)	12 (21.43%)	
GI Tract	7 (10%)	3 (5.36%)	
Skin	14 (21%)	9 (16.07%)	
Other **	9 (13%)	7 (13%)	
Unknown	3 (4%)	2 (4%)	
Clinical severity on admission			
SOFA	3.5 (2–6.8)	3 (1–6)	0.434
APACHE II score	16 (10–23)	15 (7.8–22)	0.336

Data are presented as frequencies (%) or medians with IQRs. Fisher exact test or Mann–Whitney test were used to calculate *p*-values, as appropriate. ** Other sources of sepsis include tonsillopharyngitis/lymphadenitis in 8 (6%), oral cavity in 4 (3%) and arthritis in 4 (3%).

**Table 2 ijms-26-02226-t002:** Laboratory findings on admission.

	MASLD (*n* = 68)	Non-MASLD (*n* = 56)	*p*-Value
CRP, mg/L	236 (158–313)	224 (137–293)	0.655
Procalcitonin, µg/L	1.9 (0.34–25)	1.7 (0.28–13)	0.583
Fibrinogen, g/L	6.1 (5.1–8.3)	6 (5.1–7.9)	0.762
Lactate, mmol/L	1.3 (1.1–3.2)	1.9 (1.4–2.8)	0.326
WBC, ×10^9^/L	14 (8.3–17)	14 (9.9–19)	0.757
Lymphocyte count, %	83 (77–90)	86 (81–91)	0.334
Neutrophil count, %	8.7 (3.8–13)	6.5 (3.1–11)	0.248
Neutrophil-lymphocyte ratio	9.5 (5.7–23)	13 (6.8–29)	0.368
Hemoglobin, g/L	125 (114–138)	117 (105–131)	0.087
Platelets, ×10^9^/L	202 (141–260)	211 (138–305)	0.591
Glucose, mmol/L	7.4 (6.4–9.9)	7 (5.9–9.5)	0.182
Urea, mmol/L	8 (5–13)	6.9 (5.4–13)	0.862
Creatinine, µmol/L	95 (69–175)	81 (62–138)	0.313
Sodium, mEq/L	139 (135–141)	138 (134–142)	0.522
Bilirubin, µmol/L	16 (9.8–19)	11 (9–19)	0.097
AST, IU/L	48 (23–98)	35 (21–62)	0.118
ALT, IU/L	35 (20–72)	26 (17–62)	0.282
GGT, IU/L	69 (34–130)	34 (22–69)	0.002
ALP, IU/L	71 (60–102)	79 (57–100)	0.894
LDH, IU/L	234 (187–352)	225 (181–330)	0.384
Albumins, g/L	31 (27–35)	32 (28–38)	0.133
INR	1.0 (0.96–1.2)	1.1 (0.99–1.3)	0.278
D-dimer, mg/L	2.4 (1.1–4.2)	2.7 (1.1–4.2)	0.904
Total cholesterol, mmol/L	4 (3.3–5.4)	3.7 (2.8–4.7)	0.523
LDL, mmol/L	2.3 (1.9–3.1)	2.2 (1.5–2.7)	0.155
HDL, mmol/L	0.7 (0.4–0.95)	0.8 (0.6–0.9)	0.888
Triglycerides, mmol/L	1.8 (1.4–2.8)	1.5 (1.0–2.1)	0.019

Data are presented as medians with IQRs. Mann–Whitney test was used to calculate *p*-values.

**Table 3 ijms-26-02226-t003:** Correlation analysis of sepsis severity measured by SOFA score and serum concentrations of measured cytokines and chemokines.

	SOFA Score							
	MASLD				Non-MASLD			
	Day 1		Day 5		Day 1		Day 5	
	r *	*p*-Value	r	*p*-Value	r	*p*-Value	r	*p*-Value
IL-1β	−0.05	0.33	−0.07	0.30	−0.12	0.19	−0.45	<0.01
IFN-α	0.30	0.06	0.23	0.04	0.10	0.22	−0.00	0.49
IFN-γ	−0.08	0.26	0.12	0.18	−0.25	0.03	−0.37	<0.01
TNF-α	0.13	0.16	0.03	0.41	0.01	0.46	−0.22	0.05
IL-6	0.25	0.02	0.37	<0.01	0.20	0.07	0.13	0.18
IL-8	0.25	0.01	−0.05	0.35	0.27	0.02	0.26	0.03
IL-10	0.24	0.03	0.11	0.23	0.04	0.40	−0.34	0.01
IL-12p70	−0.02	0.45	−0.08	0.38	0.05	0.40	−0.11	0.32
IL-17A	0.14	0.21	0.11	0.26	−0.13	0.24	−0.21	0.14
IL-18	0.13	0.13	0.21	0.04	0.23	0.04	−0.07	0.29
IL-23	−0.17	0.14	−0.31	0.03	−0.22	0.10	−0.44	<0.01
IL-33	−0.23	0.05	−0.33	0.01	−0.16	0.17	−0.19	0.14
IP-10	−0.00	0.49	−0.10	0.21	−0.17	0.11	−0.22	0.05
Eotaxin	−0.02	0.43	−0.27	0.01	−0.18	0.09	−0.23	0.04
TARC	−0.19	0.06	−0.15	0.13	−0.26	0.03	−0.31	0.01
MCP-1	0.21	0.04	0.07	0.28	−0.05	0.34	−0.07	0.31
MIP-1α	0.03	0.42	−0.04	0.41	0.21	0.11	−0.02	0.45
MIG	−0.13	0.15	0.00	0.47	0.13	0.17	0.06	0.34
ENA-78	−0.20	0.04	−0.19	0.06	−0.40	<0.01	−0.25	0.03
MIP-3α	0.07	0.28	0.07	0.28	0.15	0.13	0.05	0.36
GROα	0.03	0.40	0.06	0.30	−0.17	0.09	−0.19	0.08
I-TAC	−0.21	0.04	−0.11	0.20	−0.07	0.29	−0.13	0.18
MIP-1β	−0.10	0.21	0.35	<0.01	−0.03	0.40	−0.06	0.32
TGF-β1	0.07	0.31	−0.23	0.04	0.06	0.34	−0.03	0.43

* Spearman correlation analysis with corresponding r and *p*-values. Significant positive correlations are marked in orange and negative in blue.

## Data Availability

The datasets generated during and/or analyzed during the current study are available from the corresponding author upon reasonable request.

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
