# Peer review of "The Impact of Steatotic Liver Disease on Cytokine and Chemokine Kinetics During Sepsis"

_ijms, 2025, doi:10.3390/ijms26052226_

Round 1
Reviewer 1 Report
Comments and Suggestions for Authors
study explores how Metabolic Dysfunction-Associated Steatotic Liver Disease (MASLD) influences sepsis outcomes by analyzing serum cytokine and chemokine levels in patients with MASLD and sepsis. Of 124 patients, 68 had MASLD. The study found that patients with MASLD had higher concentrations of certain cytokines (e.g., IL-17A, IL-23, IL-33, CXCL10, TGF-β1) and exhibited distinct cytokine kinetics during hospitalization. Specifically, MASLD patients showed decreases in some cytokines (IL-22, IL-23, CXCL10, TGF-β1) and increases in others (IL-33, CXCL5, CXCL1) by day 5. Certain cytokine changes, particularly in TGF-β1 and CCL11, were linked to higher mortality in MASLD patients. The findings suggest MASLD may alter immune responses in sepsis, potentially affecting disease severity and outcomes.
The authors are simply looking at the effect of MASLD on serum cytokine and chemokine levels, and unfortunately, I do not find the paper highly significant.
The study suggests that distinct cytokine profiles in MASLD patients could influence sepsis outcomes, but it doesn't explore how these findings might be applied clinically. For instance, could targeting specific cytokines improve sepsis management in MASLD patients? It would be valuable to link these findings to potential therapeutic strategies or diagnostic markers.
There is no Validated method for data acquisition and the reliability of the analysis is poor. Also, it is not clear why the statistical analysis is used in the paper, and it seems to be simply to obtain significant differences.
Author Response
Study explores how Metabolic Dysfunction-Associated Steatotic Liver Disease (MASLD) influences sepsis outcomes by analyzing serum cytokine and chemokine levels in patients with MASLD and sepsis. Of 124 patients, 68 had MASLD. The study found that patients with MASLD had higher concentrations of certain cytokines (e.g., IL-17A, IL-23, IL-33, CXCL10, TGF-β1) and exhibited distinct cytokine kinetics during hospitalization. Specifically, MASLD patients showed decreases in some cytokines (IL-22, IL-23, CXCL10, TGF-β1) and increases in others (IL-33, CXCL5, CXCL1) by day 5. Certain cytokine changes, particularly in TGF-β1 and CCL11, were linked to higher mortality in MASLD patients. The findings suggest MASLD may alter immune responses in sepsis, potentially affecting disease severity and outcomes.
The authors are simply looking at the effect of MASLD on serum cytokine and chemokine levels, and unfortunately, I do not find the paper highly significant.
The study suggests that distinct cytokine profiles in MASLD patients could influence sepsis outcomes, but it doesn't explore how these findings might be applied clinically. For instance, could targeting specific cytokines improve sepsis management in MASLD patients? It would be valuable to link these findings to potential therapeutic strategies or diagnostic markers.
Authors’ response: We agree with the reviewer that our findings at this point could not be applied clinically. Our study was not designed to answer the question if targeting specific cytokines would improve sepsis management in MASLD patients, but to detect general changes in cytokine/chemokine concentrations during sepsis in MASLD patients. And our study should be viewed within this context, and this is stated in the limitations section.
There is no Validated method for data acquisition and the reliability of the analysis is poor. Also, it is not clear why the statistical analysis is used in the paper, and it seems to be simply to obtain significant differences.
Authors’ response: We thank the reviewer for this comment. Statistical methods are now described in more detail, graphs include upper and lower error bars, the usage of nonparametric tests was highlighted, as well as the lack of adjustment for multiplicity in RM-ANOVA analysis.
Reviewer 2 Report
Comments and Suggestions for Authors
Drs Vrsaljko et al report on concentrations of cyto(/chemo)kines in a cohort of 124 patients hospitalized with community-acquired. The overall sample was segregated according to having been diagnosed with metabolic dysfunction-associated steatotic liver disease (MASLD). Demographic and clinical characteristics at admission were largely similar between groups (MASLD vs non-MASLD), with notable exceptions for BMI and CAP. Cyto(/chemo)kine concentrations were compared between groups as well as over time (Day 1 to Day 5) within groups; correlations between cyto(/chem)okines and sepsis severity and steatosis grade were also assessed. Mortality was also screened for association with these markers.
This review focuses on the statistical methods used, their implementation, and reporting/interpreting those results Overall, the paper’s section detailing the statistical methods is well described and the results are presented in a clear and coherent manner through a series of tables and figures, with careful captions/legends describing the information contained in these elements. General and specific concerns are listed below, with approximate line numbers for specific points.
General:
The statistical methods section notes that both Pearson’s and Spearman’s correlation were used; however, I could only find instances where Spearman’s correlation was presented. Supplemental Table S2 and Supplemental Figure S2 both report Spearman’s correlation coefficient and those reported correlation coefficients also appear in the text. Likewise, Table 3 also relies on Spearman’s correlation. (So the text is reporting Spearman’s correlation as ‘r’ and in the supplementary information/Table 3 as ‘rho’). If there really are Pearson correlations being used, it should be noted as such; otherwise, just report in the methods that Spearman’s correlation was used (and don’t mention Pearson’s).
Similar to the above, the methods notes that t-tests may have been used to analyzes differences between groups at Day 1 and 5. Was there really ever time when t-tests were used in that manner? All details I see in the manuscript seem to rely on Mann-Whitney-Wilcoxon tests (rather than t-tests).
Table 1: The sample sizes (68 and 56) do not support reporting percentages out to 2 decimal places. When sample sizes are less than 100, please report percentages as whole numbers (e.g., 31/68 males is 46%, not 45.59%). P-values can also be given out to a maximum of 3 decimal places.
Table 1: The ‘Source of sepsis’ block has frequency counts that sum to marginal column totals of 59 and 49 (not 68 and 56), yet the percentages reported continues to use these totals (68 and 56) in formation of the percentages. At the same time a chi-square test of the 5 x 2 table of counts produces a p-value of 0.728 [not 0.0728]. Please review and correct, perhaps by giving a footnote to explain where/why the column totals deviate from what is stated in the header row.
Figure 1 & Figure 2 & Figure 3. The caption for each reads “Shown are medians with IQRs.” This cannot be accurate since there is no visualization for the 25th percentile. If the goal is to show the median and IQR, then a boxplot (central line in the box as median; upper and lower box edges 75th / 25th percentiles) would be more appropriate. Did the authors draw the bars so that a bar’s height is the median and the whisker extends upward to the 75 th percentile (or does it extend up to the maximum value or to something else)? What about showing the 25 th percentile … right now the lower limit is 0 for everything, so that can’t be the 25 th percentile and isn’t even likely to be the observed minimum in the sample. Please either redraw the graphs as boxplots to show what is claimed in the caption, or alter the description of the caption, being specific about what the bar height represents and what information is captured by the the extending whisker. In the case of Figure 3, the analysis involves an ANOVA, which relies on the means and variances (not medians with IQRs). Perhaps in Figure 3 what the authors are presenting (because of the analyses used) is the mean as the bar height and the SD as the height of the whisker extending above the bar?
It appears that Figure 3 is the only place where any sort of adjustment is made for multiple testing (i.e., Tukey’s HSD procedure used to establish significance). The authors should comment somewhere that reported p-values have not been adjusted for multiplicity. Indeed, given the large number of correlations and differences tested, it is very likely that some instances found with p<0.05 are false positives. Ideally, the authors would be able to adjust the p-values (via Holm or Benjamini-Hochberg) to give a more honest impression of which cyto(/chemo)kines have genuine differences among the many that were screened. While I understand that doing so would be a tremendous undertaking / re-analysis, the simpler solution is to make an explicit statement that no such adjustment was performed and let the readers exercise their own caution.
Lines 142–143: Currently reads as “CAP positively correlated with IFN-gamma (r=0.16), IL-23 (r=0.26) and TARC (r=0.23) on day 1; …” However, looking at Supplementary Table 2, the correlations of 0.16 [IFN-gamma] and 0.23 (TARC) appear to be on Day 5. The correlations involving CAP with those two markers on Day 1 are both small and non-significant at that time point. Also, the manuscript mentions CXCL10 in the text, but that name is not given in Supp. Table 2. It would be helpful if the authors could put the alternative names (e.g., CXCL-10 is IP-10; CCL11 is Eotaxin; etc., much like was done in Supp. Table 1) in Supplementary Table 2 to help readers find the necessary rows were the information is provided.
Author Response
Drs Vrsaljko et al report on concentrations of cyto(/chemo)kines in a cohort of 124 patients hospitalized with community-acquired. The overall sample was segregated according to having been diagnosed with metabolic dysfunction-associated steatotic liver disease (MASLD). Demographic and clinical characteristics at admission were largely similar between groups (MASLD vs non-MASLD), with notable exceptions for BMI and CAP. Cyto(/chemo)kine concentrations were compared between groups as well as over time (Day 1 to Day 5) within groups; correlations between cyto(/chem)okines and sepsis severity and steatosis grade were also assessed. Mortality was also screened for association with these markers.
This review focuses on the statistical methods used, their implementation, and reporting/interpreting those results Overall, the paper’s section detailing the statistical methods is well described and the results are presented in a clear and coherent manner through a series of tables and figures, with careful captions/legends describing the information contained in these elements. General and specific concerns are listed below, with approximate line numbers for specific points.
Authors’ responses: We thank the reviewer for the careful consideration that was given to the original version of the manuscript. We have addressed all the issues raised in the revision.
General:
The statistical methods section notes that both Pearson’s and Spearman’s correlation were used; however, I could only find instances where Spearman’s correlation was presented. Supplemental Table S2 and Supplemental Figure S2 both report Spearman’s correlation coefficient and those reported correlation coefficients also appear in the text. Likewise, Table 3 also relies on Spearman’s correlation. (So the text is reporting Spearman’s correlation as ‘r’ and in the supplementary information/Table 3 as ‘rho’). If there really are Pearson correlations being used, it should be noted as such; otherwise, just report in the methods that Spearman’s correlation was used (and don’t mention Pearson’s).
Similar to the above, the methods notes that t-tests may have been used to analyzes differences between groups at Day 1 and 5. Was there really ever time when t-tests were used in that manner? All details I see in the manuscript seem to rely on Mann-Whitney-Wilcoxon tests (rather than t-tests).
Authors’ responses: We thank the reviewer for this comments. As reviewer noted, we used non-parametric tests, and this is now more clearly stated in the manuscript.
Table 1: The sample sizes (68 and 56) do not support reporting percentages out to 2 decimal places. When sample sizes are less than 100, please report percentages as whole numbers (e.g., 31/68 males is 46%, not 45.59%). P-values can also be given out to a maximum of 3 decimal places.
Authors’ responses: We have corrected decimals and p-values.
Table 1: The ‘Source of sepsis’ block has frequency counts that sum to marginal column totals of 59 and 49 (not 68 and 56), yet the percentages reported continues to use these totals (68 and 56) in formation of the percentages. At the same time a chi-square test of the 5 x 2 table of counts produces a p-value of 0.728 [not 0.0728]. Please review and correct, perhaps by giving a footnote to explain where/why the column totals deviate from what is stated in the header row.
Authors’ responses: We thank the reviewer for noticing this mistake. In Table 1 - the “Source of sepsis” block, category “Other” was missing. This is now corrected.
Figure 1 & Figure 2 & Figure 3. The caption for each reads “Shown are medians with IQRs.” This cannot be accurate since there is no visualization for the 25th percentile. If the goal is to show the median and IQR, then a boxplot (central line in the box as median; upper and lower box edges 75th / 25th percentiles) would be more appropriate. Did the authors draw the bars so that a bar’s height is the median and the whisker extends upward to the 75 th percentile (or does it extend up to the maximum value or to something else)? What about showing the 25 th percentile … right now the lower limit is 0 for everything, so that can’t be the 25 th percentile and isn’t even likely to be the observed minimum in the sample. Please either redraw the graphs as boxplots to show what is claimed in the caption, or alter the description of the caption, being specific about what the bar height represents and what information is captured by the the extending whisker. In the case of Figure 3, the analysis involves an ANOVA, which relies on the means and variances (not medians with IQRs). Perhaps in Figure 3 what the authors are presenting (because of the analyses used) is the mean as the bar height and the SD as the height of the whisker extending above the bar?
Authors’ responses: We thank the reviewer for this comment. The whiskers now show both upper and lower IQR in Figure 1 an 2. We also clarified legend in Figure 3.
It appears that Figure 3 is the only place where any sort of adjustment is made for multiple testing (i.e., Tukey’s HSD procedure used to establish significance). The authors should comment somewhere that reported p-values have not been adjusted for multiplicity. Indeed, given the large number of correlations and differences tested, it is very likely that some instances found with p<0.05 are false positives. Ideally, the authors would be able to adjust the p-values (via Holm or Benjamini-Hochberg) to give a more honest impression of which cyto(/chemo)kines have genuine differences among the many that were screened. While I understand that doing so would be a tremendous undertaking / re-analysis, the simpler solution is to make an explicit statement that no such adjustment was performed and let the readers exercise their own caution.
Authors’ responses: We agree with the reviewer that ideally p-values should be adjusted for multiplicity. We have added this limitation in Figure 3 legend and in Methods (Statistics).
Lines 142–143: Currently reads as “CAP positively correlated with IFN-gamma (r=0.16), IL-23 (r=0.26) and TARC (r=0.23) on day 1; …” However, looking at Supplementary Table 2, the correlations of 0.16 [IFN-gamma] and 0.23 (TARC) appear to be on Day 5. The correlations involving CAP with those two markers on Day 1 are both small and non-significant at that time point. Also, the manuscript mentions CXCL10 in the text, but that name is not given in Supp. Table 2. It would be helpful if the authors could put the alternative names (e.g., CXCL-10 is IP-10; CCL11 is Eotaxin; etc., much like was done in Supp. Table 1) in Supplementary Table 2 to help readers find the necessary rows were the information is provided.
Authors’ responses: We have corrected Supplementary tables and text of the manuscript.
Round 2
Reviewer 1 Report
Comments and Suggestions for Authors
The analytical method should be validated.
Author Response
Reviewer's comment: „The analytical method should be validated".
Author's response: Chemokine and cytokine quantification method used in this manuscript (bead-based flow cytometry) is a standardised analyical methods that has been well-established in the field of flow cytometry. Specifically, we used two standardised panels (LEGENDplex Human Inflammation Panel and Proinflammatory Chemokine panel 1) that have been validated by the manufacturer of the assay (Biolegend, San Diego, CA, USA) and verified in the laboratory that performed the test (Department of Immunological and Molecular Diagnostics, University Hospital for Infectious Diseases) and has been extensively used in various research papers published by scientists from our group (co-authors of this paper). We have added this information in the text of the Methods.
- Validation of LEGENDplex panels by the manufacturer (Biolegend, San Diego, CA, USA).
1.1. LEGENDplex Human Inflammation Panel
Results of assay validation are available at: https://d1spbj2x7qk4bg.cloudfront.net/Files/Images/media_assets/pro_detail/datasheets/750000393-human-Inflammation-Panel-1-R5.pdf?v=20250203073815
(pages 22-26)
The enclosed information includes data on assay sensitivity (including LOD in assay buffer and LOD in matrix), cross-reactivity, accuracy (spike recovery), linearity of dilution, intra-assay precision, inter-assay precision, data on cytokine quantification in serum from healthy individuals, in supernatans collected after 42 hours of human PMBC cultivation in DMEM upon stimuation (by LPS, 1 μg/mL; R488, 2 μg/ mL; or with IFN-γ, 100 ng/mL) and lysate of A549 cells after 16h cultivation in DMEM and 24h stimulation with 50 ng/mL IFN-γ and 10 ng/mL TNF-α in vitro.
1.2. LEGENDplex Proinflammatory chemokine panel
Results of assay validation are available at: https://d1spbj2x7qk4bg.cloudfront.net/Files/Images/media_assets/pro_detail/datasheets/750000818-Hu-Proinflammatory-Chemokine-Panel-1-R3.pdf?v=20250203073815
(pages 21-27)
Similarly to the previous panel, the enclosed information includes data on assay sensitivity (including LOD in assay buffer and LOD in matrix), cross-reactivity, accuracy (spike recovery), linearity of dilution, intra-assay precision, inter-assay precision, data on cytokine quantification in serum and plasma from healthy individuals and cell culture supernatant.
- Laboratory verification of panels at the Department of Immunological and Molecular Diagnostics
Laboratory verification of panels was performed by comparing the concentrations of individual biological response modifiers by bead-based flow cytometry and standardised ELISA tests (IP-10 and IFN-γ) that were within ±0.5 SD.
- The assay was validated in several research models and types of samples (serum, plasma, CSF, urine) that resulted in scientific publications including:
Radmanic Matotek L, Zidovec-Lepej S, Salek N, Vince A, Papic N. The Impact of Liver Steatosis on Interleukin and Growth Factors Kinetics during Chronic Hepatitis C Treatment. J Clin Med. 2024 Aug 16;13(16):4849. doi: 10.3390/jcm13164849. PMID: 39200991; PMCID: PMC11355301.
Zidovec-Lepej S, Bodulić K, Bogdanic M, Gorenec L, Savic V, Grgic I, Sabadi D, Santini M, Radmanic Matotek L, Kucinar J, Barbic L, Zmak L, Ferenc T, Stevanovic V, Antolasic L, Milasincic L, Hruskar Z, Vujica Ferenc M, Vilibic-Cavlek T. Proinflammatory Chemokine Levels in Cerebrospinal Fluid of Patients with Neuroinvasive Flavivirus Infections. Microorganisms. 2024 Mar 26;12(4):657. doi: 10.3390/microorganisms12040657. PMID: 38674602; PMCID: PMC11052399.
Zidovec-Lepej S, Vilibic-Cavlek T, Ilic M, Gorenec L, Grgic I, Bogdanic M, Radmanic L, Ferenc T, Sabadi D, Savic V, Hruskar Z, Svitek L, Stevanovic V, Peric L, Lisnjic D, Lakoseljac D, Roncevic D, Barbic L. Quantification of Antiviral Cytokines in Serum, Cerebrospinal Fluid and Urine of Patients with Tick-Borne Encephalitis in Croatia. Vaccines (Basel). 2022 Oct 29;10(11):1825. doi: 10.3390/vaccines10111825. PMID: 36366333; PMCID: PMC9698853.
Zidovec-Lepej S, Vilibic-Cavlek T, Barbic L, Ilic M, Savic V, Tabain I, Ferenc T, Grgic I, Gorenec L, Bogdanic M, Stevanovic V, Sabadi D, Peric L, Potocnik-Hunjadi T, Dvorski E, Butigan T, Capak K, Listes E, Savini G. Antiviral Cytokine Response in Neuroinvasive and Non-Neuroinvasive West Nile Virus Infection. Viruses. 2021 Feb 22;13(2):342. doi: 10.3390/v13020342. PMID: 33671821; PMCID: PMC7927094.
Vilibic-Cavlek T, Zidovec-Lepej S, Ledina D, Knezevic S, Savic V, Tabain I, Ivic I, Slavuljica I, Bogdanic M, Grgic I, Gorenec L, Stevanovic V, Barbic L. Clinical, Virological, and Immunological Findings in Patients with Toscana Neuroinvasive Disease in Croatia: Report of Three Cases. Trop Med Infect Dis. 2020 Sep 14;5(3):144. doi: 10.3390/tropicalmed5030144. PMID: 32937866; PMCID: PMC7557803.
Reviewer 2 Report
Comments and Suggestions for Authors
Thank you for the updates/corrections to the initial manuscript. I am happy with the changes and have no further comments or concerns.
Author Response
We thank the reviewer for suggestions how to improve our manuscript.